# Three-Dimensional Graphene Composite Containing Graphene-SiO_2_ Nanoballs and Its Potential Application in Stress Sensors

**DOI:** 10.3390/nano9030438

**Published:** 2019-03-15

**Authors:** Bowei Zhao, Tai Sun, Xi Zhou, Xiangzhi Liu, Xiaoxia Li, Kai Zhou, Lianhe Dong, Dapeng Wei

**Affiliations:** 1School of Opto-Electronic Engineering, Changchun University of Science and Technology, Changchun 130022, China; bowei199396@163.com; 2Chongqing Institute of Green and Intelligent Technology, Chinese Academy of Sciences, Chongqing 400714, China; zhouxi16@mails.ucas.edu.cn (X.Z.); xiangzhiliu26@126.com (X.L.); 13996235534@163.com (X.L.); 13996234446@163.com (K.Z.)

**Keywords:** graphene, three-dimensional structure, stress sensor

## Abstract

Combining functional nanomaterials composite with three-dimensional graphene (3DG) is a promising strategy for improving the properties of stress sensors. However, it is difficult to realize stress sensors with both a wide measurement range and a high sensitivity. In this paper, graphene-SiO_2_ balls (GSB) were composed into 3DG in order to solve this problem. In detail, the GSB were prepared by chemical vapor deposition (CVD) method, and then were dispersed with graphene oxide (GO) solution to synthesize GSB-combined 3DG composite foam (GSBF) through one-step hydrothermal reduction self-assembly method. The prepared GSBF owes excellent mechanical (95% recoverable strain) and electrical conductivity (0.458 S/cm). Furthermore, it exhibits a broad sensing range (0–10 kPa) and ultrahigh sensitivity (0.14 kPa^−1^). In addition, the water droplet experiment demonstrates that GSBF is a competitive candidate of high-performance materials for stress sensors.

## 1. Introduction

Three-dimensional graphene (3DG) is a new type of aerogel, which is composed of interconnected graphene sheets to form a 3D macro-network [1,2]. It also maintains the unique properties of a two-dimensional system, including a large specific surface area, flexibility, and conduction. More notably, the porous structure of 3DG offers potential access to adjust mechanical properties and electrical conductivity by loading or modification approach [3,4]. These properties can further broaden its applications in sensing, adsorption, catalysis, and energy storage [5,6].

In recent years, various methods have been developed to prepare 3DG materials, such as freeze casting [7,8], chemical vapor deposition (CVD) [9,10], and hydrothermal methods [11,12]. However, these 3DG are easily damaged irreversibly when suffering large strain or stress, which limits the practical application greatly. Hence, their mechanical strength needs to be further enhanced. Several novel synthesis methods have been developed to prepare 3DG with excellent mechanical properties and electrical conductivity. For example, Un et al. prepared a hierarchical 3DG by a directional freezing method and high-temperature thermal treatment. The electrical conductivity of this foam can be as high as 0.5 S/cm, while its recoverable strain is only 60% [13]. Lv et al. have demonstrated a fully air-bubbled graphene foam with an extremely high sensitivity of 258 kPa^−1^, but the sensing range is limited to a stress range of 0 to 0.12 kPa [14]. Qin et al. have fabricated a rGO/PI nanocomposite using a freeze-casting strategy that shows a maximum recoverable deformation of 90% and a sensitivity of 0.18 kPa^−1^ in a stress range of 0–1.5 kPa [15]. However, in the large pressure regime (3.5–6.5 kPa), the sensitivity of rGO/PI composite is only 0.023 kPa^−1^. Hence, a 3DG material with both high sensitivity and wide sensing range is still lacking.

In general, SiO_2_ nanoparticles are ideal additives to reinforce the mechanical properties of composite materials. In addition, the addition of SiO_2_ nanoparticles always result in the reduction of the electrical conductivity. Uniform graphene coated on the SiO_2_ nanoparticles by a CVD process could maintain electrical conductivity well while improving the mechanical properties [16,17]. In this paper, a one-step hydrothermal reduction self-assembly method was developed to synthesize three-dimensional graphene composite foam (GSBF) combined with GSB. This GSBF composite simultaneously owns excellent mechanical properties (95% strain), a wide sensing range (0–30 kPa), and a high sensitivity (0.14 kPa^−1^). Additionally, the GSBF has very low density of 7.7 mg/cm^3^ and a particularly high electrical conductivity of 0.458 S/cm. Moreover, the GSBF can be used as a stress sensor for detecting frequencies of water droplets. The results suggest that the GSBF composite has exciting application prospects as a stress sensor in the field of real-time monitoring.

## 2. Materials and Methods

### 2.1. Synthesis of GSB

As shown in Figure 1a, GSB was prepared by chemical vapor deposition (CVD) [16,17]. First, SiO_2_ nanoparticles of diameters from 20 to 30 nm were placed in a furnace, and the air in the tube was purged by ventilating argon gas through it. Then, the temperature was raised to 1000 °C in 50 min while keeping the argon gas flowing in the tube at 200 sccm. A CH_4_ gas flow of 25 sccm was used as the carbon source when the temperature reached 1000 °C, and the argon gas flow was cut at the same time. After 20 min, the CH_4_ gas was turned off and the argon gas flow was restarted. Finally, the furnace was naturally cooled to room temperature under argon gas atmosphere and GSB was obtained.

### 2.2. Preparation of GO

The Figure 1b shows that GO was synthesized by a modified Hummer’s method [18,19]. Firstly, 12 g of graphite and 6 g of NaNO_3_ were added to 220 mL of concentrated sulfuric acid under stirring in a flask immersed in an ice-water bath. Then, 30 g of KMnO_4_ was added slowly, and the mixture was stirred at 30 °C for 2 h. Next, 460 mL of distilled water was added, and the mixture was stirred for another 30 min at 95 °C. Finally, 940 mL of distilled water and 30 ml of H_2_O_2_ (5%) were added to the mixture to terminate the reaction, and the color of the solution turned from dark-brown to yellow. The generated solid graphite oxide was separated from the solution by centrifugation. After being washed, and dried under vacuum, pure GO was obtained.

### 2.3. Preparation of GSBF Composite

The GSBF composite was prepared through a one-step hydrothermal reduction self-assembly method [20]. The synthesis process of GSBF composite were shown in Figure 1c,d. The obtained GSB were added to 20 mL of a 2 mg/mL GO solution in a weight ratio of GO:GSB (10:0 to 10:3), and 0.12 g sodium dodecyl benzene sulfonate was added at the same time to ensure that GSB was soluble in mixed solution. Then the mixed solution was ultrasonically dispersed for 30 min. When the mixed solution had been dispersed well, 50 µl of ethylenediamine (EDA) and 80 µl of sodium borate solution (SBS) were added to the dispersion. After being uniformly mixed, the solution was transferred to a Teflon reactor and incubated at 120 °C for 6 h to obtain a GSBF hydrogel. The residual solvents were removed from GSBF hydrogel by immersing it in 20% ethanol solution for 24 h. Next, the hydrogel was frozen in a low temperature freezer at −78 °C for 20 h, and then transferred to a freeze dryer and stayed there for 20 h to obtain an aerogel of the GSBF composite. Finally, the GSBF aerogel was placed in a tube furnace and treated at 800 °C for 1 h under argon atmosphere to obtain the GSBF composite material.

### 2.4. Characterization

The microscopic morphology of GSBF (10:1) was obtained by a scanning electron microscope (SEM, JSM-7800F, JEOL, Tokyo, Japan) and transmission electron microscope (TEM, Talos F200S, Thermo Fisher Scientific, Shanghai, China). X-ray diffraction (XRD) patterns were required using a diffractometer (X‘Pert3 Powder, PANalytical B.V., Almelo, The Netherlands) with a Cu Kα radiation source (λ = 1.5406 Å). X-ray photoelectron spectroscopy (XPS) (ESCALAB 250Xi, Thermo Fisher Scientific, Shanghai, China) was carried out on a spectrometer with monochromatic Al Kα (1486.71 eV) X-ray radiation (15 kV and 10 mA). Raman spectroscopy (InVia Reflex Renishaw, Gloucestershire, UK) was used to obtain the Raman spectra of materials. Thermo Gravimetric Analysis (TGA, DSC1, Mettler Toledo, Zurich, Switzerland) was used to measure the carbon content of GSB.

### 2.5. Mechanics and Electrical Measurement

The mechanical properties of GSBF materials were tested by repeated stretching experimental instruments (SBA-50S, Jinshidun, Suzhou, China). The compression strain rate was 3 mm/min, and the recovery speed was 3 mm/min. The electrical conductivity of the GSBF was measured by a four-probe DC method (MCP-T700, Mitsubishi, Hong Kong, China). Electrical information was collected by a SourceMeter (2450, Keithley, Beaverton, OR, USA).

## 3. Results and Discussion

### 3.1. Microscopic Characterization

It can be found that as the C content increased, the electrical conduction of the GSBF improved while the mechanical properties deteriorated. The TGA results of GSB with different C contents are shown in Appendix A. In order to obtain both good electrical conductivity and mechanical properties at the same time, we choose the GSB synthesized with a C content of 51% in 20 min as the optimum candidate. Pictures of GSBF samples prepared with different proportions of the constituents are shown in the inset of Figure 2a (10:0 to 10-3 from left to right), and the photograph (Appendix A) of GSBF at the ratio of 10:1 shows its ultralight density. It was observed that the shape of the GSBF integrated with GSB (10:1 to 10:3) was more regular than that of the graphene foam (GF, whose ratio is 10:0). The recoverable strain, sensitivity, sensing range, Young’s modulus, and electrical conductivity of GSBF composites with different ratios are listed in Table 1.

Comparing GSBF (10:1–10:3) with GF (10:0) (GF-to-GSBF ratio of 10:0), significant improvements in the sensing range and mechanical strength were observed. The mechanical strength comparison of GSBF (10:0 to 10:3) is shown in Appendix A. This greatly increases the range of applications for sensors based on GSBF materials. The reason behind these improvements may be the existence of various interactions between GSB and GO sheets, including Van der Waals forces, π-π stacking, physisorption, and hydrophobic interaction. It should be noted that the surface of GSB was completely coated with graphene, and the particle size of each GSB was very small (about 40 nm), which could provide better π-π stacking interactions during the reduction self-assembly process compared to other metal nanoparticles [21,22]. However, as the GSB content increased from 10:1 to 10:3, the adhesion of GSB to the graphene sheet led to an increasing degree of stacking and negative effect in the three-dimensional interconnected structure between the graphene sheets. This explains the decrease observed in the recoverable strain and Young’s modulus of GSBF as the ratio decreased from 10:1 to 10:3.

As shown in Figure 2a,b (Appendix A), GSBF had a continuous 3D interconnected porous structure with many GSBs attached to the graphene nanosheets uniformly. The high uniformity displayed here is because during the mixing process, GSB can be well dispersed in GO solution, forming stable mixed suspensions with the assistance of sodium dodecyl benzene sulfonate. In addition to this, GO sheets have large lateral dimension and an array of hydrophobic, π-conjugated nanopatches in their basal plane. As a result, in the mixed suspensions, GO sheets were able to capture the GSB with multiple adhesion sites by various interactions (Van der Waals forces, π-π stacking and physisorption) between GO sheets and GSB. Consequently, the GSBF was uniformly attached with GSB after the reduction self-assembly. It can be seen from Figure 2c that the GSB attached to the graphene nanosheets had a diameter of about 40 nm. The high-magnification TEM (Transmission Electron Microscope) images of GSB are shown in Figure 2d and Appendix A and reveal that GSB was successfully coated by several layers of graphene in the CVD process.

### 3.2. Structural Analysis

Figure 3a shows the XRD patterns of GO, GF, and GSBF (XRD pattern of GSB is shown in Appendix A). The GO sample showed a sharp peak at around 2θ = 11.06°, which means that GO has a comparatively large interlayer spacing of 0.80 nm. This is because many oxygen-containing functional groups, such as hydroxyl, epoxy, and carboxyl groups, are attached to the GO sheet layer. When GO was transformed into GSBF, the sharp peaks of GSBF and GF shifted to 2θ = 25.05°, and the interlayer distance was reduced to 0.36 nm [23]. This result indicated that after hydrothermal self-assembly, the number of oxygen-containing functional groups was significantly reduced. The peak at 2θ = 20.37° of SiO_2_ in the GSBF curve revealed that the addition of GSB was successful and did not affect the hydrothermal self-assembly process between graphene sheets [17]. To further demonstrate the structural change of GSBF, the Raman spectra are shown in Figure 3d which shows that the peaks of GO, GF, and GSBF were all located at 1349 cm^−1^ and 1587 cm^−1^ corresponding to the D and G bands, respectively [24]. Upon completion of the reduction process, ID/IG increased from 0.9 to 1.19 after GO was transformed into GSBF, indicating that a certain number of structural defects had been introduced into graphene during the reduction process. For the GSBF sample, the band at 968 cm^−1^ was assigned to SiO_2_ [16,17], revealing that GSB had been composited into the GSBF composites (the Raman spectrum of GSB is shown in Appendix A).

As shown in Figure 3b,c,e,f, XPS was used to characterize the elemental compositions of GSBF and GO. Figure 3b is the survey spectrum of GSBF which shows the peaks at 102, 284.8, and 532.5 eV were assigned to the characteristic peaks of Si 2p, C 1s, and O 1s [11,17], respectively. Figure 3e,f are the fitting curves of GSBF and the original GO. A comparison of the two graphs shows that the sharp peaks of the C-O (286.6 eV) and C = O (288.4 eV) bonds of GO were much higher than those of GSBF, which means that GO was highly reduced into graphene after the reduction self-assembly process [25]. Figure 3g–k are the energy-dispersive spectroscopy (EDS) maps of GSBF. It can be clearly seen that the GSBF composite is consisted of three elements: C, O, and Si, where C was assigned to graphene, and O and Si corresponds to SiO_2_ nanoparticles. From the layers of C, O, and Si in Figure 3h, it was confirmed that the surface of SiO_2_ was completely covered by graphene.

### 3.3. Stress-Strain Performance of GSBF

To demonstrate the mechanical properties of GSBF composites, especially the recoverable compressive deformation capability, a series of mechanical tests with GSBF composites (10:1) were conducted. As shown in Figure 4a, strain is used as the independent variable and compressive stress is used as the dependent variable. The stress–strain curves in Figure 4a and Appendix A show that the GSBF composite can recover to the original position after the release of each strain from 5% to 95%. The high recoverability was attributed to the porous interconnected network structure of the GSBF and the strength enhancement of three-dimensional graphene walls brought by GSB. When compressive stress was applied to GSBF, the decrease in volume caused the air in the GSBF network structure to be expelled. However, the volume quickly returned to its original size as air re-entered into the GSBF during unloading compressive stress due to the excellent mechanical properties of the GSBF three-dimensional network structure [13].

In addition, the stress–strain curve obtained during loading (Figure 4b) can be divided into two different linear phases: the first linear region ε < 60% (ε is the strain), and the second linear region where 60% < ε < 95%. In the first region of ε < 60%, the pores of the GSBF are gradually compressed during the process of increasing strain, which causes the graphene sheets to be continuously bent and stacked. However, because of the existence of pores, the stacking speed of graphene sheets is much slower than the bending speed. The stress is mainly provided by the curved graphene sheets. Therefore, the stress increases slowly with the increase of strain in this region. But in the second region of 60 < ε < 95%, as the strain gradually increases, most of the air in the pores is squeezed out, resulting in a large number of graphene sheets stacked together. Hence, during the subsequent compression process, more stress is required to produce the same deformation. So, the curve in second region is rising more sharply than the first region. Furthermore, the unloading curves in Figure 4a almost returned to their original position after each deformation, indicating that the GSBF composite did not exhibit plastic deformation. This pronounced deformation behavior is usually observed in porous interconnected network structures [33,34]. The stress sensitivity of GSBF is defined as the slope of the curve in Figure 4c, which is given by the following formula:S = δ (ΔI/I_0_)/δP(1)
ΔI/I_0_ = (I − I_0_)/I_0_(2)
where I_0_ is the current without stress applied, I is the current value corresponding to the applied stress, and P is the applied stress. According to Figure 4c, the GSBF composite material showed a broad sensing range from 0 to 10 kPa after the introduction of GSB, and the sensitivity reached up to 0.14 kPa^−1^. In the range of 15–30 kPa, the composite material had a sensitivity of 0.03 kPa^−1^. Compared to sensors made from other composite materials, sensors proposed here overcome the problems of ultra-high sensitivity with low sensing range (229.8 kPa^−1^ in 0–0.12 kPa) [14], or low sensitivity with a high sensing range (0.023 kPa^−1^ in 0–200 kPa) [35]. The comparison between sensors based on other materials reported in recent years and our sensor is shown in Figure 4d and Appendix A.

### 3.4. Stress Sensor Characterization of GSBF

The frequency response of a stress sensor is one of the most important characteristics. Therefore, the frequency response of GSBF composites was studied at different frequencies with 20% strain. The results in Figure 5a show that the output curve of the GSBF sensor remained distinguishable as the frequency increased from 0.01 Hz to 0.1 Hz, and the peak intensities of different curves were at identical levels. This result demonstrates that the GSBF material has a fast and stable response as a stress sensor. To further demonstrate the stability of the GSBF sensor, its stress-current stability was investigated for 1300 cycles at 11 kPa. As can be seen from the graph in Figure 5b,c the stress–current curve remained stable for 1300 cycles, and there was almost no peak fluctuation of the curve. Only a slight descent in the curve can be observed at its bottom because as the number of compressions increased, some of the conductive paths of GSBF underwent irreversible damage which induced an increase in the initial resistance. Then, the initial current gradually decreased to a stable state against the compression recovery cycle. During the external stress loading and unloading process in Figure 5d, the current values of the GSBF foam were 0.14 A, 0.26 A, and 0.38 A for 10%, 30%, and 50% strain, respectively. This shows that the increase in the current was almost linear with the applied strain. Figure 4e presents the current-voltage curve of the GSBF composite under different compressive stresses. As seen from the curve, the current shows a consistent linear increase with the voltage at the same stress, indicating that the GSBF sample had stable electrical characteristics. In addition, the electric delay time of the samples was measured, and shown in Figure 5f that the GSBF sensor had response time of about 260 ms.

### 3.5. Water-Dripping Test

Additionally, a water-dripping test was performed on the GSBF sensor in order to demonstrate the application prospects of GSBF in real-time monitoring. As shown in Figure 1d, the bottom and top of the sensor were covered with polyethylene terephthalate (PET) films and two copper wires were used as the electrodes connected with GSBF by silver paste. In this experiment (Figure 6a), water droplets falling at different frequencies were used to simulate raindrops. Each water droplet corresponds to a peak of the curve in Figure 6b, which indicates that the difference between dripping at slow and fast frequencies could be simultaneously detected by the GSBF sensor. The results above demonstrate that the three-dimensional interconnected structure enforced by GSB endowed the GSBF with excellent mechanical and electrical properties, implying that the GSBF sensor has excellent application prospects as a stress sensor in the field of real-time monitoring.

## 4. Conclusions

In summary, a novel GSB-combined three-dimensional graphene material composite was developed by a one-step hydrothermal reduction self-assembly method. The GSBF material with a three-dimensional porous interconnected structure exhibits an ultra-low density (7.7 mg/cm^3^), a high electrical conductivity (0.458 S/cm) and a maximum recoverable compressive stress of 112 kPa at 95% compressive strain. Most importantly, the stress sensor based on the GSBF material simultaneously meets the requirements of high sensitivity and a wide working range. The sensitivity is 0.14 kPa^−1^ in the stress range of 0–10 kPa and 0.03 kPa^−1^ in the stress range of 15–30 kPa. While maintaining a high sensitivity (0.14 kPa^−1^), the sensing range (0–10 kPa) was several times greater than that of other reported materials. In addition, the GSBF sensor is able to detect the different frequencies of water dripping, which demonstrates that the GSBF sensor has great potential in the field of real-time stress sensor monitoring.

## Figures and Tables

**Figure 1 nanomaterials-09-00438-f001:**
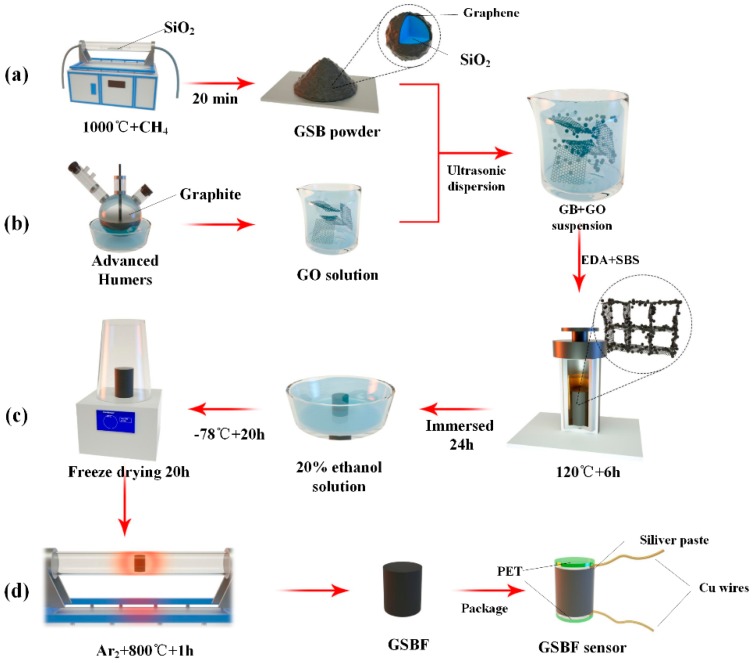
The schematic illustration of synthesis process of three-dimensional graphene composite foam (GSBF) composite and GSBF sensor. (**a**) synthesis of graphene-SiO2 balls (GSB). (**b**) preparation of graphene oxide (GO) solution. (**c**,**d**) the preparation process of GSBF and GSBF sensor.

**Figure 2 nanomaterials-09-00438-f002:**
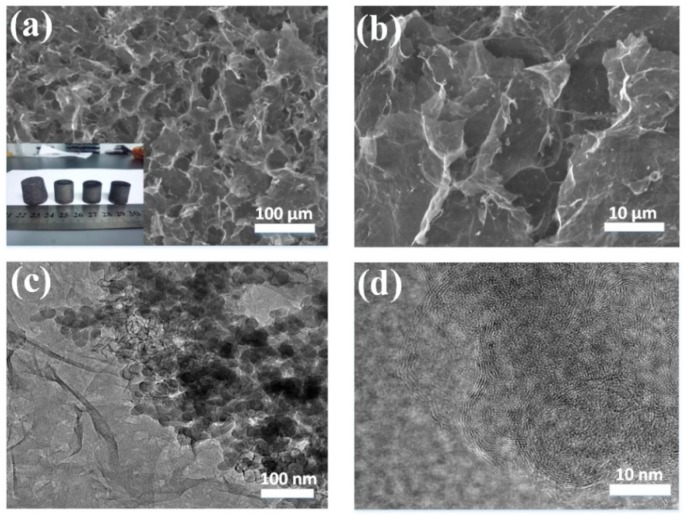
(**a**,**b**) Scanning electron microscopy (SEM) images of three-dimensional graphene composite foam (GSBF) (10:1), insert of (**a**) is the photograph of GSBF with different ratios of graphene oxide (GO) to graphene-SiO2 balls (GSBs) (the ratio of GO to GSB is 10:0, 10:1, 10:2, 10:3 from left to right). (**c**) Low-magnification transmission electron microscopy (TEM) images of GSBF. (**d**) High magnification TEM images of GSBF.

**Figure 3 nanomaterials-09-00438-f003:**
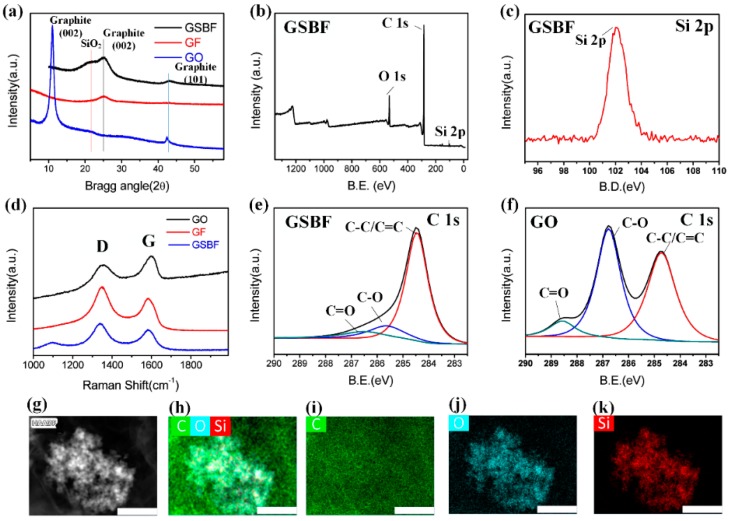
(**a**) The X-ray diffraction (XRD) spectra of three-dimensional graphene composite foam (GSBF), graphene foam (GF), and graphene oxide (GO). (**b**) X-ray photoelectron spectroscopy (XPS) spectrum of GSBF. (**c**) XPS spectrum of Si 2p. (**d**) Raman spectra of GSBF, GF and GO. (**e**,**f**) Curve fitting of the C 1 s spectra of GSBF and GO samples. (**g**–**k**) EDS mapping graph of GSBF elements; the scale is 500 nm.

**Figure 4 nanomaterials-09-00438-f004:**
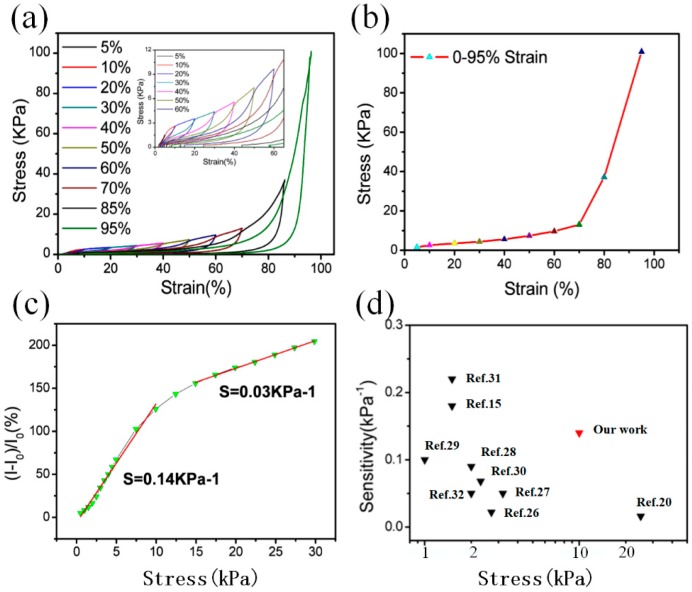
(**a**) The stress-strain curves of the three-dimensional graphene composite foam (GSBF) composite with 0 to 95% strain, inside of (**a**) is an enlarge figure (**b**) The stress–strain curve of GSBF (graphene oxide (GO): graphene-SiO2 balls (GSB) ratio of 10:1) with 0–95% strain. (**c**) Resistance response at different frequencies under 30% strain. (**d**) The comparison of sensors in previous reports and this work [15,20,26,27,28,29,30,31,32].

**Figure 5 nanomaterials-09-00438-f005:**
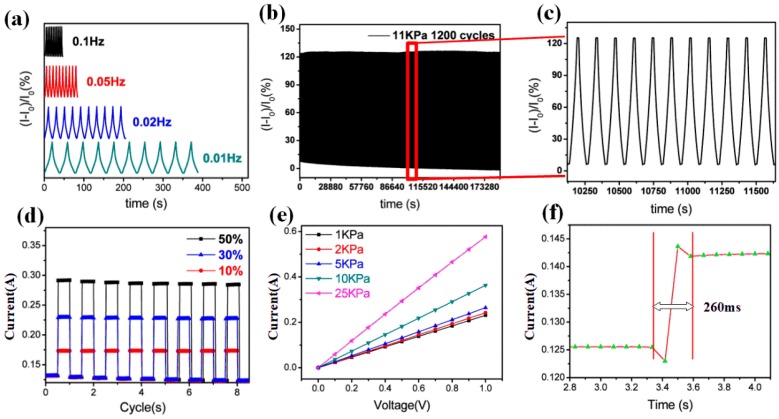
(**a**) Current response at different frequencies under 11 kPa. (**b**) Stress-current curve for 1300 cycles at 11 kPa. (**c**) Partial view of Figure 4e. (**d**) Current curves at different strains. (**e**) Current–Voltage curves at different stresses. (**f**) Response time of three-dimensional graphene composite foam (GSBF) to the applied stress.

**Figure 6 nanomaterials-09-00438-f006:**
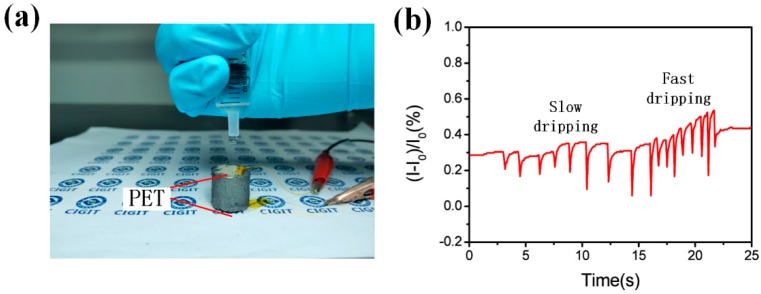
(**a**) The test of dripping water droplets on the GSBF sensor. (**b**) The curve of water-dripping test.

**Table 1 nanomaterials-09-00438-t001:** The mechanical properties and electrical conductivity of three-dimensional graphene composite foam (GSBF) with different ratios.

No.	Ratio (GO:GSB)	Recoverable Strain (%)	Sensitivity (kPa^−1^)	Sensing Range (kPa)	Young’s Modulus (kPa)	Electrical Conductivity (S/cm)
**1**	10:0	>95	0.4	0–2	48	0.1
**2**	10:1	95	0.14	0–10	112	0.184
**3**	10:2	90	0.12	0–8	119	0.186
**4**	10:3	80	0.08	0–5	98	0.188

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
