# Peer review of "Three-Dimensional Graphene Composite Containing Graphene-SiO2 Nanoballs and Its Potential Application in Stress Sensors"

_nanomaterials, 2019, doi:10.3390/nano9030438_

Round 1
Reviewer 1 Report
The paper describes fabrication, characterization and a stress sensor application of 3D graphene composite material doped with SiO2 nanoballs. To the best of my knowledge, the work is novel and original. The title of the paper is appropriate. The abstract outlines the aim and the approach of the work and points out the most important results. In the introduction, the work is put into proper context to prior publications on graphene composite materials and stress sensors made thereof. In the Materials and Methods section, the synthesis of GSB and the preparation of GO and GSBF are described in sufficient detail. The SEM, XRD, XPS, Raman and TGA instruments used for characterization are briefly introduced. In the Results and Discussion section (and in the supplement), a whole lot of characterization experiments are described. The section needs some more structuring, e.g. by subheadings. An application sample for raindrop counting is given. In the conclusion, the approach and the main findings are summarized.
I suggest to revise and improve the paper, considering the following comments:
1) Abstract: I found several language-related errors in the abstract (see attached annotated manuscript). The acronym “GO” needs to be explained. In addition, there are two almost identical sentences at the end. Please re-write.
2) Introduction: I missed a comparison to other (non-graphene) stress sensors. The introduction needs to be revised with respect to language, too.
3): Results and Discussion. This section is very long. Its structure is difficult to grasp. I suggest to add sub-headings, e.g. “3.1. Microscopic characterization”, “3.2. Structural analysis”, “3.3. Stress-Strain performance”, “3.4. Stress sensor characterization”, “3.5. Water dripping test”, or similar. Maybe it would be useful to move Figure 1 and the first paragraph of Section 3 to “Materials and Methods”, as they are related to the preparation protocol.
4) Figure 3: please consistently use “GSBF” (some graphs are labeled “GBF”).
5) Figure 4 (a): The figure is too small. I could not distinguish the ten curves. Please enlarge the figure or reduce the amount of information contained.
6) Are there other envisioned applications beyond a water-dripping test?
7) Several acronyms need to be explained at first usage, e.g. GO, GF, and PET.
I marked several typos and suggestions for language improvement in the attached annotated manuscript.

Author Response
Response to the Reviewer 1’s comments
Comments:
The paper describes fabrication, characterization and a stress sensor application of 3D graphene composite material doped with SiO2 nanoballs. To the best of my knowledge, the work is novel and original. The title of the paper is appropriate. The abstract outlines the aim and the approach of the work and points out the most important results. In the introduction, the work is put into proper context to prior publications on graphene composite materials and stress sensors made thereof. In the Materials and Methods section, the synthesis of GSB and the preparation of GO and GSBF are described in sufficient detail. The SEM, XRD, XPS, Raman and TGA instruments used for characterization are briefly introduced. In the Results and Discussion section (and in the supplement), a whole lot of characterization experiments are described. The section needs some more structuring, e.g. by subheadings. An application sample for raindrop counting is given. In the conclusion, the approach and the main findings are summarized.
1) Abstract: I found several language-related errors in the abstract (see attached annotated manuscript). The acronym “GO” needs to be explained. In addition, there are two almost identical sentences at the end. Please re-write.
Thank you for your advices and I had corrected these problems of The acronym “GO” and deleted the duplicate sentence.
2) Introduction: I missed a comparison to other (non-graphene) stress sensors. The introduction needs to be revised with respect to language, too.
The center of this article is a three-dimensional graphene composite, so it focuses on devices of the same type of material. To highlight the superiority of composite materials in this paper.
3) Results and Discussion: This section is very long. Its structure is difficult to grasp. I suggest to add sub-headings, e.g. “3.1. Microscopic characterization”, “3.2. Structural analysis”, “3.3. Stress-Strain performance”, “3.4. Stress sensor characterization”, “3.5. Water dripping test”, or similar. Maybe it would be useful to move Figure 1 and the first paragraph of Section 3 to “Materials and Methods”, as they are related to the preparation protocol.
Your suggestion is very helpful for my paper. Added with these sub-headings, my paper has surely become more organized and easier to comprehend.
4) Figure 3: please consistently use “GSBF” (some graphs are labeled “GBF”).
It has been corrected from “GSB” to “GSBF”.
5) Figure 4 (a): The figure is too small. I could not distinguish the ten curves. Please enlarge the figure or reduce the amount of information contained.
The figure has been enlarged to ensure that the curves can be distinguished more easily.
6) Are there other envisioned applications beyond a water-dripping test?
Yes. Apart from the water-dripping test, GSBF sensor also is able to detect different breath states such as: normal breath, deep breath and fast breath.
7) Several acronyms need to be explained at first usage, e.g. GO, GF, and PET.
The explanations of the first usage (GO, GF, PET) have been added.
Reviewer 2 Report
The authors, Zhao et al., were reported a work of 3D composite materials of C-coated SiO2/rGO for the compressive pressure sensor. The composite material is somehow interesting, and the composite sensor performed good towards a large range of pressrure (1-10 KPa). However, the work is not inside the mechanism of the formation of composite structures and some parts could be improved. The reviewer therefore recommends for consideration of publication after a major revision.
Comments to the authors
Te title of the work should be changed, especially a term of “…doped with functional nanoballs” doesn’t make sence.
The introduction could be improved, and cited some update refences which are reval to the work, such as: 3D graphene foam-reinforced polymer composites – A review, Carbon 2018, 135, 52-71; Recent advances in sensing applications of grapheme assemblies and their composites. Adv. Funct. Mater. 2017, 27,1702891, Electrical percolation in graphene–polymer composites. 2D Mater. 5 (2018) 032003
There is some research literature related to the work could be cited, typically, Graphene Coating of Silicon Nanoparticles with CO2‐Enhanced Chemical Vapor Deposition. Small 2015, 12, 658
English could be editing, even in the abstract there are few grammartical and type errors can be seen.
The interface in composite materials, and the dispersion of filler (i.e. C-coated SiO2 particles) are highly important, and that should be interpretation in the research work, this is somehow the authors should be discussed in the paper
The conductivity measurement should be described in the experiment section.
I think the term “GSBF doped GSB” should be changed throught out the work, probadly using “combined or intergrated/blended”. The authors use few times “electronic properties (0.485 S/cm)”, this is “electrical conductivity”, make sure that should be corrected.
Authors reported that using CVD to prepared graphene-coated SiO2 with CH4 gas (experiment section). The review think that somehow a strange here, without reducing gas (e.g., H2) or mild oxidant (e.g., CO2) there might be not formed graphene coating, but Carbon coated, and might be SiC phase could be formed
Author Response
Response to the Reviewer 2’s comments
Comments:
The authors, Zhao et al., were reported a work of 3D composite materials of C-coated SiO2/rGO for the compressive pressure sensor. The composite material is somehow interesting, and the composite sensor performed good towards a large range of pressrure (1-10 KPa). However, the work is not inside the mechanism of the formation of composite structures and some parts could be improved. The reviewer therefore recommends for consideration of publication after a major revision.
1) The title of the work should be changed, especially a term of “…doped with functional nanoballs” doesn’t make sence.
The title has been changed as “A Three-dimensional Graphene Composite Material Doped with Graphene-SiO2 Nanoballs and Its Potential Application in Stress Sensors”
2) The introduction could be improved, and cited some update refences which are reval to the work, such as:“3D graphene foamreinforced polymer composites – A review, Carbon 2018, 135, 5271”;“Recent advances in sensing applications of grapheme assemblies and their composites. Adv. Funct. Mater. 2017, 27,1702891”; “Electrical percolation in graphene-polymer composites. 2D Mater. 5 (2018) 032003”;There is some research literature related to the work could be cited, typically, “Graphene Coating of Silicon Nanoparticles with CO2Enhanced Chemical Vapor Deposition. Small 2015, 12, 658”.
The above refences are very useful for my article introduction, and I have cited them in my paper.
3) English could be editing, even in the abstract there are few grammartical and type errors can be seen.
Thank you for your suggestion. The English has been edited again.
4) The interface in composite materials, and the dispersion of filler (i.e. C-coated SiO2 particles) are highly important, and that should be interpretation in the research work, this is somehow the authors should be discussed in the paper。
The analysis of dispersion was discussed in part 3.1. In the mixing process , GSB can be well dispersed in GO suspensions, forming stable mixed suspensions with the assistance of sodium dodecyl benzene sulfonate. The hydrophobic basal plane and hydrophilic edges of GO sheet makes it act as a surfactant. Besides, GO sheet has large lateral dimension and an array of hydrophobic, π-conjugated nanopatches in their basal plane. As a result, in the mixed suspensions, GO sheets are able to capture the GSB with multiple adhesion sites by various interactions between GO sheets and GSB, such as van der Waals interaction, physisorption, hydrophobic or π–π interaction.
5) The conductivity measurement should be described in the experiment section.
The measurement for conductivity has been added in the part 2.5.
6) think the term “GSBF doped GSB” should be changed throught out the work, probadly using “combined or intergratedended”. The authors use few times “electronic properties (0.485 S/cm)”, this is “electrical conductivity”, make sure that should be corrected.
These are very proper advices and I have taken them by substituting “combined” with “doped” and “electronic properties ” with “electrical conductivity”.
7) Authors reported that using CVD to prepared graphenecoated SiO2 with CH4 gas (experiment section).The review think that somehow a strange here, without reducing gas (e.g., H2) or mild oxidant (e.g., CO2) there might be not formed graphene coating, but Carbon coated, and might be SiC phase could be formed.
In the CVD progress CH4 gas was fed into a furnace in the presence of SiO2 nanoparticles at 1000 °C. At this temperature, CH4 is decomposed to generate hydrogen atoms, which can subsequently reduce SiO2 to SiOx (x < 2). OH− is also simultaneously produced via the following reaction:
SiO2 + CH4 → SiOx + OH- + 3H+ + Carbon(Graphene)
The SiOx provides catalytic sites for graphene growth and OH- serves as a mild oxidant to facilitate the graphitic carbon formation toward graphene.
Round 2
Reviewer 1 Report
The manuscript has been improved with respect to content and readability. Most of my suggestions were implemented. However, almost
all typos and language errors that I marked in the annotated manuscript
attached to my first review report were ignored. The paper still needs a thorough revision with respect to language. Section number 3.3 is duplicate.

Author Response
Thank for your marks about the typos and suggestions for language improvement. After careful consideration of your comments, I have made corrections to the errors you have marked.
Reviewer 2 Report
1. The title ‘Three-dimensional Graphene Composite Material Doped with Graphene-SiO2 Nanoballs and Its Potential Application in Stress Sensors’, suggested a change to “A Three-dimensional Graphene Composite Containing Graphene-SiO2 Nanoballs and Its Potential Application in Stress Sensors”
2. The electronic conductivity should be changed to “electrical conductivity” in the abstract, authors should make sure to change throughout the MS, there are quite a few, i.e. line 108. The term ‘conductivity’, for example, in line 116 and in conclusion, should be replaced with the clear term “electrical conductivity”
3. The strain sensing versus deformation (<12%, and="">12%) could be explained, as reference work: J.F. Feller. Carbon 108, 2016, 450-460
Author Response
1.The title ‘Three-dimensional Graphene Composite Material Doped with Graphene-SiO2 Nanoballs and Its Potential Application in Stress Sensors’, suggested a change to “A Three-dimensional Graphene Composite Containing Graphene-SiO2 Nanoballs and Its Potential Application in Stress Sensors”
Thank you for your advice. The title has been change to “A Three-dimensional Graphene Composite Containing Graphene-SiO2 Nanoballs and Its Potential Application in Stress Sensors”.
2.The electronic conductivity should be changed to “electrical conductivity” in the abstract, authors should make sure to change throughout the MS, there are quite a few, i.e. line 108. The term ‘conductivity’, for example, in line 116 and in conclusion, should be replaced with the clear term “electrical conductivity”
Thank for your advice. All the errors have been corrected to “electrical conductivity”.
3. The strain sensing versus deformation (<12%, and="">12%) could be explained, as reference work: J.F. Feller. Carbon 108, 2016, 450-460
Thank for your suggestion. After reading the article, I have added the explains about the strain sensing versus deformation. Besides, I have changed the regions from “ε < 12%, 12% <ε < 60% and 60% <ε < 95%” to “ε < 60% and 60% <ε < 95%”.
In the first region of ε<60%, the pores of the GSBF are gradually compressed during the process of increasing strain, which causes the graphene sheets to be continuously bent and stacked. However, because of the existence of pores, the stacking speed of graphene sheets is much slower than the bending speed. The stress is mainly provided by the curved graphene sheets. Therefore, the stress increases slowly with the increase of strain in this region. But in the second region of 60<< span="">ε<95%, as the strain gradually increases, most of the air in the pores is squeezed out, resulting in a large number of graphene sheets stacked together. Hence, during the subsequent compression process, more stress is required to produce the same deformation. So the curve in second region is rising more sharply than the first region.